# Enhancing the Efficacy of Radiation Therapy by Photochemical Internalization of Fibrin-Hydrogel-Delivered Bleomycin

**DOI:** 10.3390/cancers16234029

**Published:** 2024-11-30

**Authors:** Sophia Renee Laurel, Keya Gupta, Jane Nguyen, Akhil Chandekar, Justin Le, Kristian Berg, Henry Hirschberg

**Affiliations:** 1Beckman Laser Institute, University of California, Irvine, CA 92617, USA; slaurel@uci.edu (S.R.L.);; 2Department of Radiation Biology, The Norwegian Radium Hospital, Oslo University Hospital, Montebello, N-0310 Oslo, Norway; kristian.berg@rr-research.no

**Keywords:** fibrin glue, radiation sensitizer, photosensitizer, photochemical internalization, hydrogel

## Abstract

Glioblastoma multiforme (GBM) are highly invasive brain tumors, and despite surgery and post-operative radio/chemotherapy, GBM survival rates are poor, with less than 5% of patients remaining alive after 5 years. The efficacy of chemotherapy and radiotherapy (RT) is greatly limited by the blood–brain barrier on the one hand and normal brain tissue toxicity, which limits the allowable RT dose, on the other. Here, we present an anti-cancer drug-loaded hydrogel as a promising strategy for bypassing the blood–brain barrier (BBB). The hydrogel is loaded with a radiation-sensitizing drug that is first activated by the light-based technology photochemical internalization, which greatly enhances the anti-cancer effects of RT.

## 1. Introduction

Despite postoperative chemo- and radiotherapy, almost 100% of glioma patients suffer a post-treatment recurrence of their tumors with 80–90% around the margins of the surgical resection cavity [1,2,3]. The efficacy of chemo- and radiotherapy (RT) is greatly limited by the blood–brain barrier on the one hand, whereas normal brain tissue toxicity limits the RT dose allowable. Achieving local tumor control, therefore, inevitably fails. Strategies for augmenting the effectiveness of combined chemotherapy and RT would therefore be of substantial value.

One method of increasing the efficacy of RT is the use of a radiation-sensitizing agent (RS) that selectively enhances cell killing from irradiation in tumor cells [4,5,6]. Unfortunately, RS clinical trials for GBM, thus far, have been disappointing [7,8,9]. In the reported clinical trials, RS was given intravenously, and therefore, the blood–brain barrier (BBB) limited the concentration of these agents at the tumor site. Due to their intrinsic toxicity, the drug dose administered systemically has proven inadequate. The local delivery of therapeutic agents by implanted slow-release hydrogels at the tumor site or in the tumor resection cavity has the potential to bypass the BBB as well as increasing drug efficacy [10,11].

One form of hydrogel, fibrin glue (FG), has several characteristics, such as proven excellent biocompatibility, high drug loading capacity, sustained drug release, and ease of forming, which make it well-suited for direct drug delivery. FG loaded with chemotherapeutic drugs or radiation sensitizers has, therefore, been researched as a localized controlled release vehicle [12,13].

Hydrophilic drugs have a significantly greater loading ability into hydrogels compared to hydrophobic drugs, but they do not easily penetrate the lipid plasma membranes. They are actively transported into cells by endocytosis and remain trapped in the endosome, limiting their ability to reach their intended intracellular target [14,15].

One very effective method to promote endosomal escape is photochemical internalization (PCI) [16]. PCI utilizes the principles of photodynamic therapy (PDT) by employing specialized photosensitizers, like Aluminum phthalocyanine desulphonated (AlPcS_2a_) and meso-tetraphenyl chlorin desulphonated (TPCS2a), that preferentially accumulate in the endosome membrane. Light irradiation ruptures the endosomal membrane, allowing the released agent to exert its full biological activity, as opposed to degradation by lysosomal hydrolases following endo-lysosomal fusion [17,18,19].

PCI has been demonstrated to significantly enhance the efficacy of the anti-cancer agent bleomycin (BLM-PCI) [20,21,22,23]. In particular, BLM-PCI before external-beam radiotherapy has been shown to improve local control in a mouse sarcoma model compared to BLM or radiation acting alone [24].

The aim of the present in vitro research was to evaluate the ability of PCI of both free and FG-released drug to enhance the effectiveness of BLM as a radiation sensitizer. Multicell three-dimensional tumor spheroids formed from glioma tumor cells were used as an in vitro model.

## 2. Materials and Methods

### 2.1. Cells

The F98 rat glioma cell line was acquired from the American Type Culture Collection, while the BT_4_C cell line, originating from fetal rat brain cells exposed to ethyl-nitrosourea, was provided by Dr. Dag Sorensen of the University in Oslo, Norway. Both lines were cultured in Dulbecco’s modified Eagle’s medium (DMEM) with high glucose (Life Technologies Corp., Carlsbad, CA, USA), enriched with 2% FBS, 25 mM HEPES (pH 7.4), and antibiotics (100 U/mL penicillin, 100 µg/mL streptomycin) under 5% CO_2_ at 37 °C.

### 2.2. Chemicals

Aluminum phthalocyanine disulfonate (AlPcS_2a_), used as the photosensitizer, was acquired from Frontier Scientific, Inc. (Newark, DE, USA), while bleomycin (BLM), utilized as a chemotherapeutic agent, was purchased from Sigma Aldrich (St. Louis, MO, USA).

### 2.3. Fibrin Glue and Drug Harvest

Fibrin glue, acquired from EMD Millipore Calbiochem, was prepared by combining fibrinogen and thrombin in a 1:1 ratio, with BLM incorporated into the thrombin component. In a 24-well plate, 0.2 mL of BLM-thrombin solution was added to 0.2 mL fibrinogen, and was allowed to gel for 20 min at 37 °C. Following this, the wells were rinsed twice to remove any unbound drug, and 1.6 mL of drug-free culture medium was added. After 48 h, the supernatant containing released BLM was collected for subsequent experimental use.

### 2.4. Spheroid Formation

F98 and BT_4_C spheroids were formed from the centrifugation of 96-well round-bottomed plates (Corning Inc., Corning, NY, USA) containing 2.5 × 10^3^ F98 or BT_4_C cells in 100 μL of culture media within each well [16]. Centrifugation occurred at 500× *g* for 10 min to group the cells into a disk. Incubation of the plates occurred at 37 °C in 5% CO_2_ for 24 h prior to receiving treatment. The cells typically formed uniform three-dimensional spheroids, approximately 0.2 mm in diameter. Spheroid growth was monitored by measuring spheroid diameters with a digital inverted microscope (BZ-X810 Fluorescence Microscope. KEYENCE, Itasca, IL, USA). Spheroid volumes were then calculated assuming a perfect sphere.

### 2.5. Radiation and PCI Treatment

The basic experimental protocol for the FG-released drug is shown in Figure 1. It consisted of five arms: non-treatment controls (NTC), (2) RT only, (3) BLM^FG^-PCI only, (4) BLM^FG^ +RT only, and (5) BLM^FG^-PCI + RT.

Spheroids were incubated with 0.1 mL of free 0.03 or 0.06 μg/mL AlPcS_2a_ with either 0–4.8 μg/mL free BLM or BLM harvest from FG (BLM^FG^). BLM or BLM^FG^ was added one hour after AlPcS_2a_ was added to the spheroid cultures. The spheroids received light treatment of λ = 670 nm, at an irradiance of 2.0 mW/cm^2^, then received RT of 1.02 Gy/min at a 50 cm beam distance immediately after or delayed BLM-PCI application. The RT was administered in an X-Rad 320 cabinet irradiator (Precision X-Ray Irradiation, Madison, CT, USA) for varying time intervals corresponding to different radiation doses. The plates returned to incubation after treatment. Each experimental arm throughout 3 independent experiments consisted of 8–16 spheroids. The spheroids were monitored for up to 14 days of incubation, post-treatment, as the culture mediums in the wells were exchanged after 7 days.

### 2.6. Statistical Analysis

Data analysis and graphing were carried out using Microsoft Excel. Throughout the study, mean values and standard deviations were consistently employed. Statistical significance was analyzed using both Student’s and Welch’s *t*-tests, with *p*-values below 0.05 indicating that there are significant differences between values. The following equation was applied to evaluate whether the FG-RS + RT effect was synergistic, antagonistic, or additive:α = (SF^a^ × SF^b^)/(SF^ab^).

SF^a^, growth following RT only. SF^b^, growth following BLM-PCI only. SF^ab^, growth following BLM-PCI + RT. α > 1, synergistic. α < 1, antagonistic. α = 1, additive.

## 3. Results

### 3.1. Effects on Spheroid Growth of RT and BLM-PCI as a Single Treatment

The effects on F98 or BT_4_C spheroids, exposed to increasing doses of RT (0–20 Gy), are shown in Figure 2a.

As seen in Figure 2a, RT as a single treatment allowed spheroids to usually reach 90–100% of NTC volume after 14 days in culture for all radiation doses up to 10 Gy. In contrast, radiation doses of 15 and 20 Gy completely inhibited spheroid growth.

In order to establish relevant BLM concentrations to be used in subsequent BLM-PCI procedures, experiments were performed using free drug over a range of increasing BLM concentrations from 0 to 4.8 μg/mL with 0.06 μg/mL of free AlPcS_2a_ and light treatment, 0.96 J/cm^2^. The results for both cell lines are shown in Figure 2b. BLM concentrations of 1.2 μg/mL and above as a single PCI treatment provided significant spheroid growth inhibition, which if used in combination with RT could potentially mask the additional effects of RT. For this reason, BLM concentrations of either 0.3 or 0.6 μg/mL were used in all the following experiments.

### 3.2. RT Effects on Spheroid Growth by BLM and BLM-PCI as Free Drug

BLM, as a free drug, included four groups: (1) NTC, (2) RT, (3) BLM + RT, and (4) BLM-PCI + RT. Figure 3a displays the effects of RT (8 Gy) on the growth of F98 and BT_4_C spheroid volume combined with BLM or BLM-PCI as a free drug. AlPcS_2a_ concentration was 0.06 μg/mL, and light radiance of 0.96 J/cm^2^ was used for all PCI experiments.

The kinetics of F98 and BT_4_C spheroid growth pattern, following NTC, RT, BLM + RT, and BLM-PCI + RT, are illustrated in Figure 3a,b. RT only (8 Gy) resulted in a significant delay in spheroid growth, but growth continued, and spheroid volume was equal to NTC volume after 14 days in culture. As seen in Figure 3a,b, RT alone (8 Gy) did produce a growth delay compared to NTC with RT-treated spheroids reaching 56% of NTC final volume on day 9 compared to 92% for control spheroids. A similar growth delay was also observed for both cell lines at RT = 10 Gy, with spheroids reaching 49% of NTC final volume at day nine. 

The ability of BLM to act as a radiation sensitizer (BLM + RT) was significant only at the highest BLM concentration of 0.6 μg/mL, with the spheroids reaching 66% of control values. In contrast, a significant inhibiting effect was seen with RT in combination with increasing BLM-PCI (BLM-PCI + RT) at both BLM concentrations tested. At a BLM-PCI concentration of 0.6 μg/mL, the treated spheroid volume was 15% of control values after 14 days of growth. Similar but less pronounced results were obtained on the cell line BT_4_C, (Figure 3b) with the spheroid volume of 18% of NTC volume at the highest BLM concentration.

### 3.3. RT Effects on Spheroid Growth by BLM^FG^ and BLM^FG^-PCI as FG-Released Drugs

The basic experimental setup is pictured in Figure 1. The optimal time for drug harvest was determined by fluorescence emission spectroscopy and reported in a previous study [13]. The FG was loaded with 1.2 μg of BLM in 0.4 mL of FG and overlaid with 1.6 mL of medium. Previous results demonstrated that although BLM was slowly released over the entire 72 h time interval tested, the difference between the BLM concentration after 48 and 72 h was not significant. In all subsequent experiments, supernatants harvested after 48 h were used. BLM^FG^ as a released drug included the same four groups as used for the free drug: (1) NTC, (2) RT, (3) BLM^FG^ + RT, and (4) BLM^FG^-PCI + RT. Loading FG with 1.5 μg of BLM was performed as described in the Materials and Methods section. The AlPcS_2a_ concentration was 0.06 μg/mL, and a light radiance of 0.96 J/cm^2^ was used for all PCI experiments. As seen in Figure 4a for F98 and Figure 4b for BT_4_C spheroids, BLM^FG^-PCI combined with 8 Gy RT showed significant growth-inhibiting effects compared to BLM^FG^ + RT as a single treatment. At the BLM^FG^-PCI dilutions of 2:1 and 1:1 combined with 8 Gy, RT resulted in spheroid volumes of 41% and 17% for F98 spheroids and 39% and 16% of control volumes for BT_4_C spheroids, respectively.

### 3.4. Simultaneous Release of Both BLM and AlPcS_2a_

In the above-described experiments, BLM^FG^ was combined with a free photosensitizer (AlPcS_2a_) to allow the results to be directly compared to the results obtained using both the free drug and photosensitizer. This protocol, however, does not completely represent expected in vivo conditions, where both photosensitizer and drug will be simultaneously and continually released. Experiments were therefore performed where both BLM (0.6, 1.2, 2.4 μg) and AlPcS_2a_ (0.12 μg) were loaded into the FG. The results shown in Figure 5a (F98) and Figure 5b (BT_4_C) clearly indicate that simultaneous FG release of both drug and photosensitizer-mediated PCI (BLM^FG^-PCI + RT) can induce significant spheroid growth inhibition at all the drug loading dosages used.

### 3.5. Comparison of RT and Light Radiation Dose on the Effects of BLM-PCI + RT

The basic hypothesis being tested in this study is that a significantly lower RT dose is required when combined with BLM-PCI than that needed for RT alone to obtain an equivalent efficacy. Experiments were performed over an RT dose range from 0 to 15 Gy with and without BLM-PCI. AlPcS_2a_ concentration was 0.06 μg/mL, BLM concentration was 0.6 μg/mL (both as a free drug), and light radiance was 0.96 J/cm^2^. The results are shown in Figure 6a for F98 spheroids. An RT dose of 15 Gy was necessary to significantly inhibit spheroid growth down to 15% of the NTC spheroid volume. For an equivalent effect, only approximately 9 Gy was required when combined with BLM-PCI at the experimental values tested.

The effects of light radiance levels of BLM-PCI with and without RT are shown in Figure 6b. AlPcS_2a_ concentration was 0.06 μg/mL, BLM concentration was 0.6 μg/mL (both as free drug), and light radiance was in a range of 0–1.68 J/cm^2^. The degree of growth inhibition increased, as expected, with increasing radiance both for the BLM-PCI and the BLM-PCI + RT groups. The addition of RT greatly increased the toxic effects of BLM-PCI at all the radiance levels tested compared to BLM-PCI alone (*p* < 0.05).

### 3.6. Effects of Treatment Delay Between BLM^FG^-PCI and RT

Experiments were performed comparing RT delivered either immediately after or delayed 24 h after BLM^F^*^G^*-PCI. F98 cells were used in both groups. FG was loaded with 2 μg of BLM, and then supernatants were harvested after 48 h. The AlPcS_2a_ concentration was 0.06 μg/mL, and a light radiance of 0.96 J/cm^2^ and RT of 8 Gy were used. The results for serial dilutions of supernatant (SN) are shown in Figure 7. Compared to BLM + RT, significant inhibition (*p* < 0.05) of spheroid growth was demonstrated for both the immediate and delayed BLM^FG^-PCI and RT groups. As shown in the figure, the effects of immediate BLM^FG^-PCI before RT were slightly greater than those obtained with a 24 h delay, but the difference at all the dilutions tested was not significant (*p* > 0.05).

Since the addition of either BLM or BLM-PCI to RT is a technique that relies on the combination of drug and RT exposure, the resultant toxicity should show more than an additive effect of the single modalities. The degree of synergism was calculated both for free BLM + RT and BLM-PCI + RT and for BLM^FG^ and BLM^FG^-PCI + RT for F98 spheroids using the formula described in the Materials and Methods. The data used for the calculation of the α values shown in Table 1 were derived from the experiment using F98 cells, 0.6 μg/mL BLM, 0.96 J/cm^2^, and a supernatant dilution of 1:1. As evident from the calculated α values shown in Table 1, BLM + RT and BLM^FG^ + RT demonstrated marginal synergy at all the radiation doses evaluated. In contrast, a significant and equivalent synergistic effect (α > 1) was obtained for both free and FG released drug combined with PCI, particularly at radiation levels of 8 and 10 Gy. Similar results were also obtained using the BT_4_C cell line (Table 1).

## 4. Discussion

Since radiation therapy forms one of the main modalities for the treatment of GBM, improving its efficacy while reducing its neurotoxicity is an important goal. One direction, mainly through advances in high-precision technologies as well as a better understanding of side effects, has achieved significant advances in local dose escalation without undue toxicities [25].

The use of radiation sensitizers, which selectively enhance radiation toxicity on tumor cells while exhibiting limited single-agent toxicity towards normal tissue, represents a second method to widen the therapeutic index of RT. Injectable hydrogel direct delivery systems have been reported for radiation enhancers [26]. This approach forms the basis for the experimental results presented here. The combination of hydrogels as direct delivery systems for radiation sensitizers with light-activated PCI produces a highly targeted site and a temporal-specific treatment modality for increasing the therapeutic index of RT. The surgically targeted placement of the radiation sensitizer in a slow-release vehicle allows for high drug concentration at the local site (the resection cavity) while reducing systemic side effects by bypassing the BBB.

Additionally, the rapid attenuation of light in the brain confines the PCI drug activation effect to a limited area, since the effect is localized to the illuminated area.

In a previous study, we demonstrated that the drug 5FU, both as a free drug and released from FG, could act as a radiation sensitizer, significantly enhancing the effects of RT [27]. This was not the case for BLM, as is clearly shown in Figure 3 for the free drug and Figure 4 for the FG-released drug. Even at relatively high BLM doses, the ability of BLM to enhance the effects of RT was limited. This is in all probability due to its entering cells by endocytosis and its subsequent entrapment.

In contrast, BLM-PCI greatly increased the ability of RT to inhibit spheroid growth, also shown in Figure 3, Figure 4 and Figure 6. PCI is known to promote endosomal escape, and once BLM is released in the cell cytosol, it quickly diffuses into the nucleus, where it has a significant effect, causing multiple double DNA strand breaks per molecule [28,29]. Ionizing radiation also produces DNA double-stranded breaks, which in turn triggers a cascade of enzymatic processes to attempt DNA repair. Our interpretation of the clear synergistic effects, as shown in Table 1, between BLM-PCI and RT is that the DNA damage caused by both modalities overwhelms the ability of the cell to repair this damage, pushing the cell into apoptosis.

In the experimental protocol used here, the drug-containing FG layer was 0.4 mL in volume while the overlaying supernatant was 1.6 mL, which totals 2 mL. Assuming that after an incubation interval of 48 h, an equilibrium drug concentration will be reached between the FG layer and the surrounding culture medium. For example, 1.2 μg of BLM loaded into the FG will, after equilibrium is reached, acquire a BLM concentration of 0.6 μg/mL in the supernatant. Adding the supernatant undiluted into the spheroid wells will, therefore, approximate the 0.6 μg/mL concentration used in the free drug experiments. This allows comparisons of the efficacy of free versus FG-released drug. The results indicate that BLM released from FG is as active as free drug and does not appear to be degraded. Additionally, comparing the α values, shown in Table 1, for free versus FG-released BLM corroborates the finding that the drug is released in an active form.

A similar calculation would also apply to the FG-released photosensitizer used in the simultaneous release experiment shown in Figure 5a,b.

The only previous report on the ability of PCI to enhance the effects of RT has shown that introducing a delay interval between BLM-PCI and RT did not alter the treatment efficacy but did significantly reduce normal tissue damage [24]. The results shown in Figure 7 also demonstrate no significant differences between immediate and delayed BLM^FG^-PCI.

### In Vivo Translation

In order to allow for wound healing, postoperative external beam radiation and chemotherapy are delayed several weeks, allowing for disease progression to occur untreated. To obtain equivalent efficacy, a significantly lower RT dose is required when combined with BLM-PCI compared to what is required for RT alone. This, in turn, allows treatment to be initiated shortly after surgical resection and during the time interval of drug release. Several methods for both light and RT administration are available. A proposed translation to a rat animal GBM model is shown in Figure 8.

One method for implementing light administration is the temporary implantation in the FG of fiberoptic catheters. The FG is semitransparent and would therefore act as a light diffuser, assuring even light delivery.

External beam RT approaches such as stereotactic radiotherapy, stereotactic radiosurgery, or CyberKnife would be well suited to deliver targeted therapy in this interval [30,31]. Another promising method is Flash radiotherapy, characterized by the delivery of ultra-high radiation doses (>40 Gy/s) in a very short interval. Flash RT has shown considerable potential for improving cancer treatment by minimizing damage to surrounding healthy tissues while effectively targeting tumors [32,33].

## 5. Conclusions

The results of the present study show that active non-degraded BLM was released from FG layers (Table 1). Although BLM^FG^ did not prove to be an efficient radiation sensitizer, BLM^FG^-PCI greatly increased the efficacy of RT. A time delay of 24 h between BLM^FG^-PCI and RT did not significantly alter the results compared to those obtained from experiments with a minimal interval between BLM^FG^-PCI and RT. To obtain equivalent RT efficacy, a significantly lower RT dose was required when combined with BLM^FG^-PCI than what was required for RT as a single treatment. The in vitro results reported here form the basis for translation to in vivo animal experiments and eventually patient treatment protocols as FG is widely clinically approved.

## Figures and Tables

**Figure 1 cancers-16-04029-f001:**
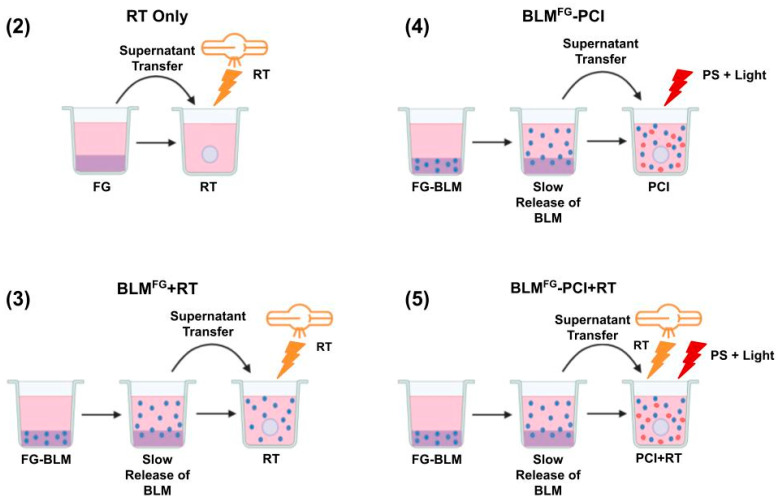
Experimental protocol. Supernatants harvested from BLM-loaded FG or non-loaded fibrin glue coated in media at varying time intervals were transferred to spheroid cultures in multi-well plates. Five experimental arms: non-treatment controls, **(2)** RT only, **(3)** BLM^FG^ +RT, **(4)** BLM^FG^ (supernatant harvested from BLM-loaded FG), **(5)** BLM^FG^ +PCI + RT.

**Figure 2 cancers-16-04029-f002:**
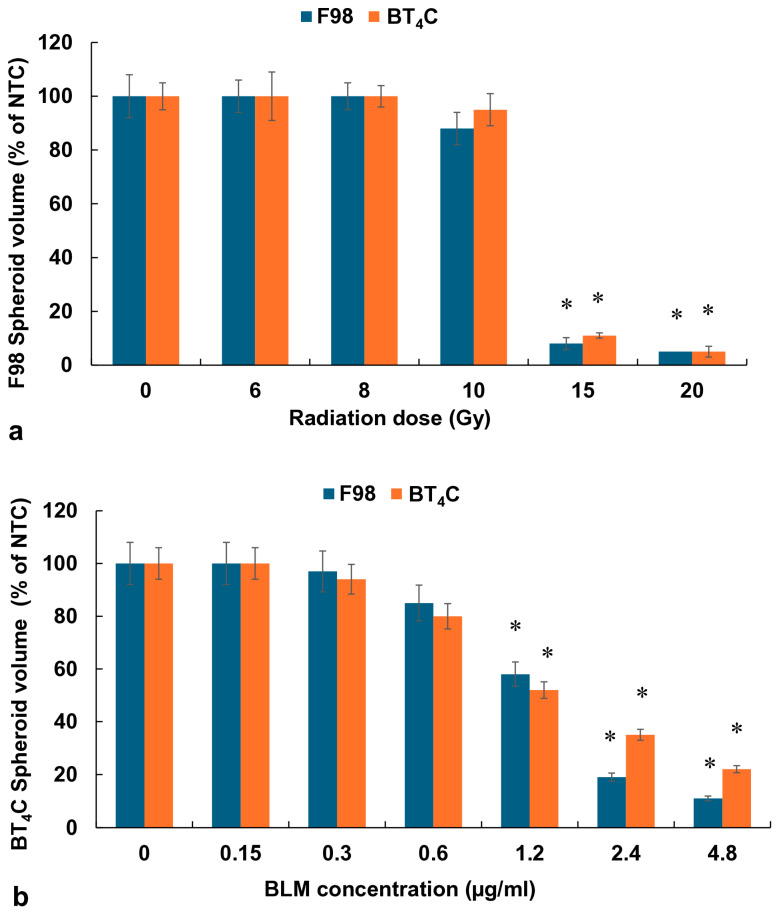
(**a**) RT effect on growth of F98 and BT_4_C spheroids over a range of radiation doses, 0–20 Gy. (**b**) Effect of BLM-PCI at varying BLM concentrations on F98 and BT_4_C spheroid growth. Data points are the average volumes of spheroids after 14 days in culture, represented as a % of the non-treatment control spheroid volumes. Error bars indicate standard deviation, and * represents significant differences (*p* < 0.05) when compared to controls.

**Figure 3 cancers-16-04029-f003:**
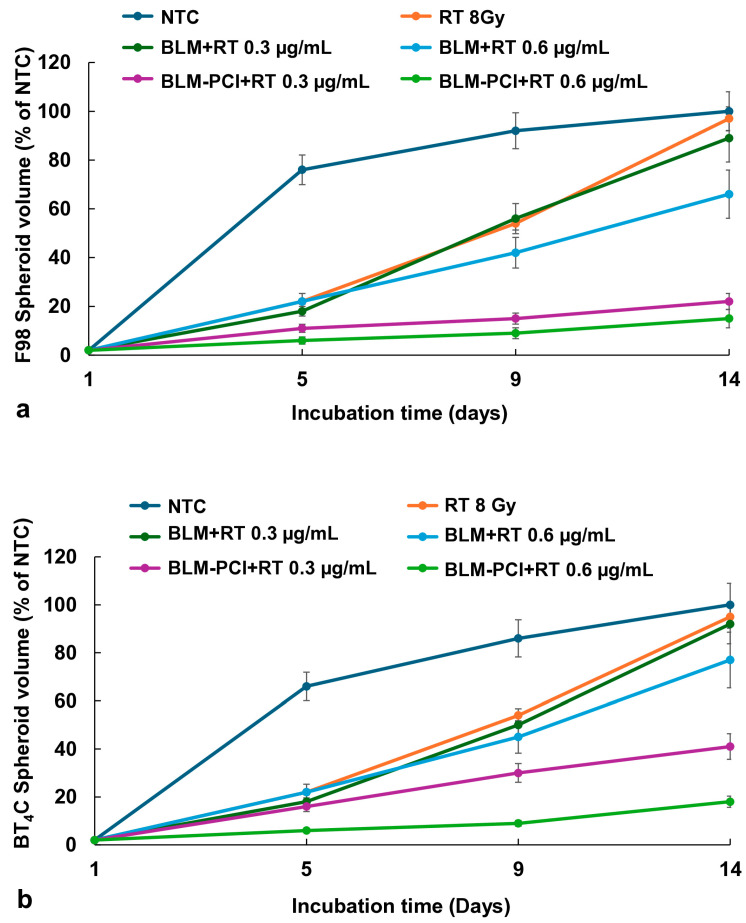
(**a**) Kinetics of the F98 spheroid growth pattern after exposure to RT of 8 Gy. (**b**) Kinetics of the BT_4_C spheroid growth pattern after exposure to RT of 8 Gy. Spheroids were treated with BLM at 0.3 or 0.6 μg/mL, combined with AlPcS_2a_ at 0.06 μg/mL, and light, 0.96 J/cm^2^. Data points are the average spheroid volumes after 14 days in culture, from 2 independent experiments, represented as a % of the non-treatment controls. Error bars indicate standard deviation.

**Figure 4 cancers-16-04029-f004:**
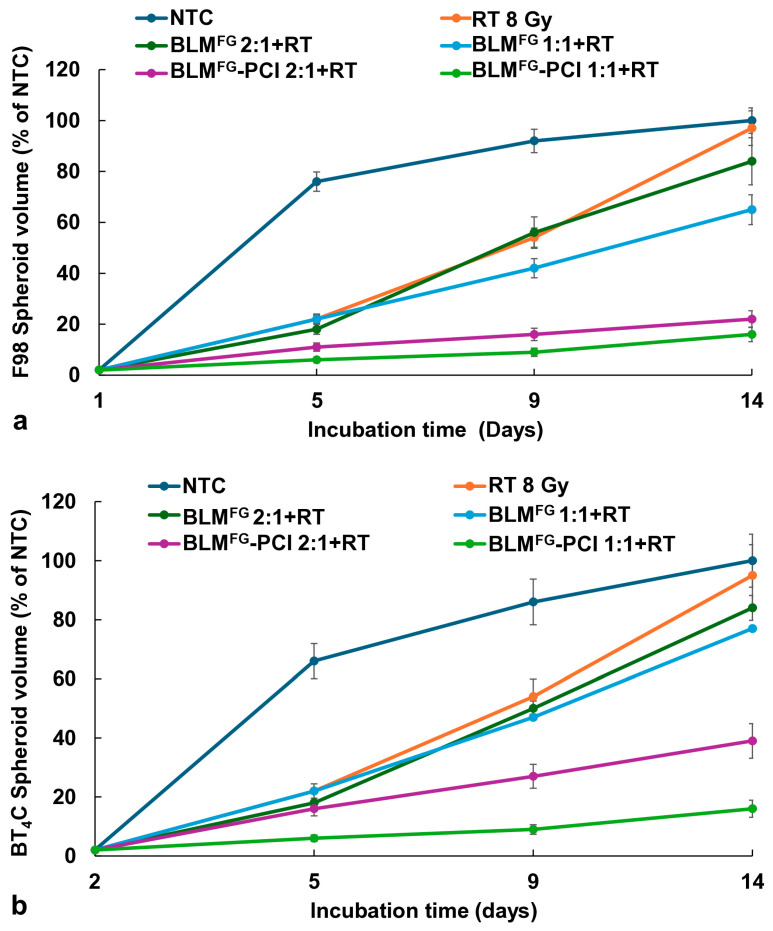
(**a**) The effect of varying dilutions of BLM^FG^-PCI on the growth kinetics of the F98 spheroid growth pattern combined with RT of 8 Gy. (**b**) Kinetics of the BT_4_C spheroid growth pattern after varying dilutions of BLM^FG^-PCI combined with RT of 8 Gy. Spheroids were treated with BLM^FG^, combined with AlPcS_2a_ at 0.06 μg/mL and light. Data points are the average volumes of spheroids from 3 independent experiments after 14 days in culture, represented as a % of the non-treatment control spheroid volumes. Error bars indicate standard deviation.

**Figure 5 cancers-16-04029-f005:**
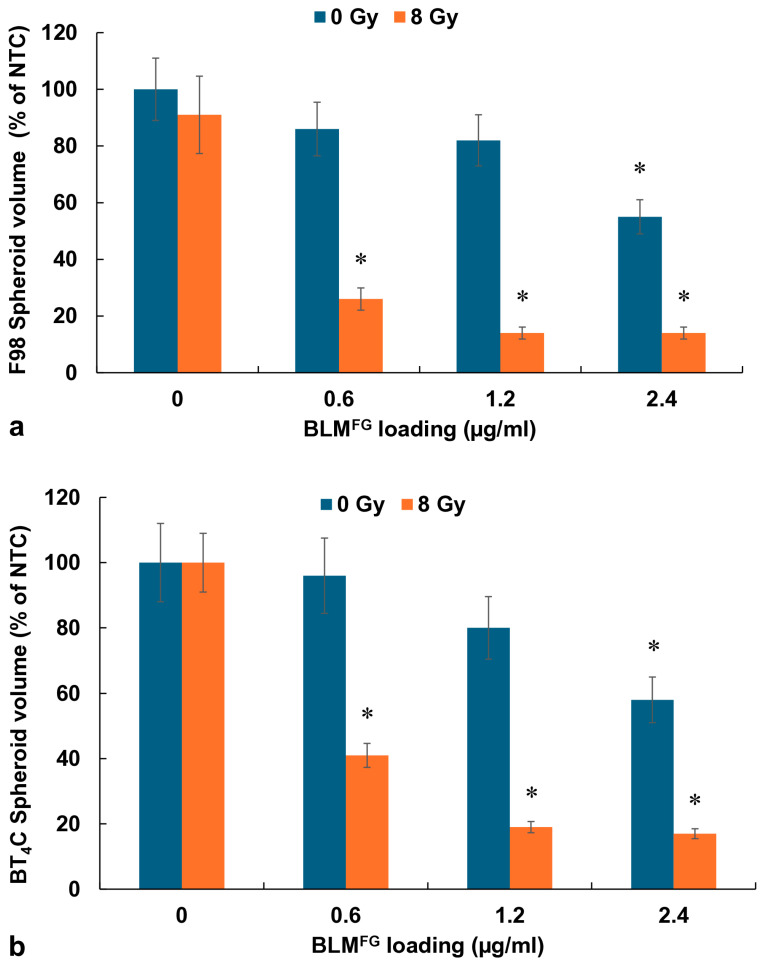
(**a**) Effect of increased concentration of BLM loading in FG, with 0–2.4 ug/mL on F98 spheroids that received no RT or received 8 Gy of RT. (**b**) BT_4_C spheroids with increasing concentration of FG-released BLM and given no RT or 8 Gy of RT. The average volumes of F98 spheroids after 14 days in culture, represented as a % of the non-treatment control spheroid volumes. Error bars indicate standard deviation, and * represents significant differences (*p* < 0.05) when compared to controls.

**Figure 6 cancers-16-04029-f006:**
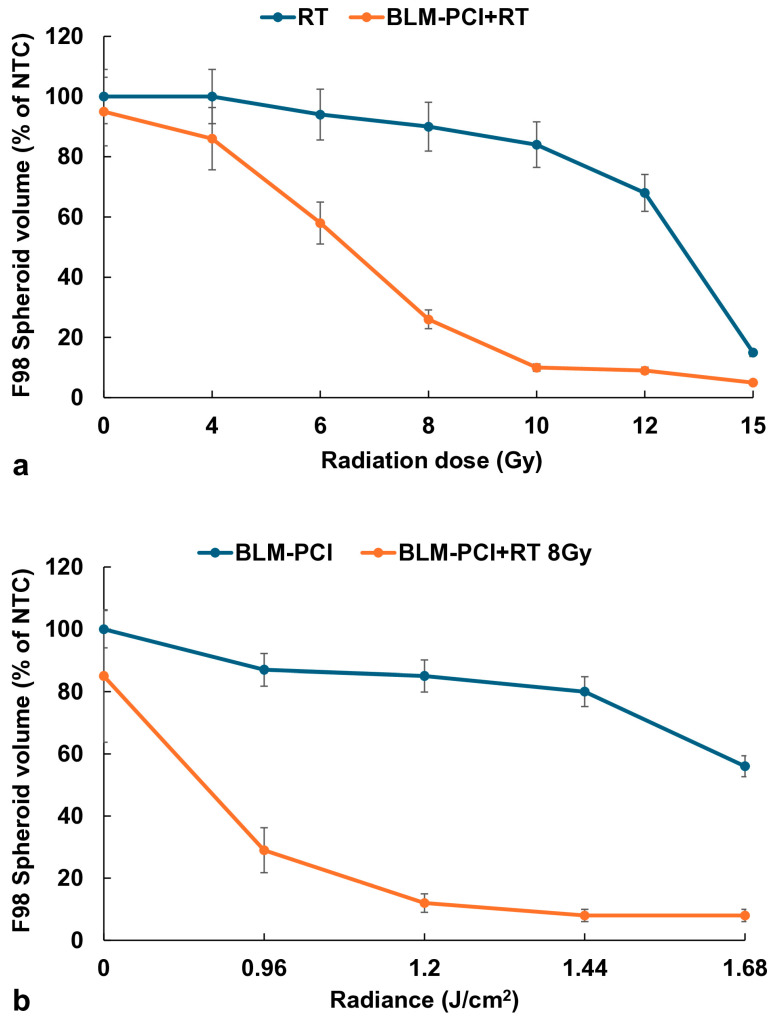
(**a**) Effects of varying RT doses with and without BLM-PCI on F98 spheroids, 0–15 Gy. (**b**) Effects of light radiance levels from BLM-PCI with and without RT, 0–1.68 J/cm^2^. The average volumes of F98 spheroids after 14 days in culture, represented as a % of the non-treatment control spheroid volumes. Error bars indicate standard deviation.

**Figure 7 cancers-16-04029-f007:**
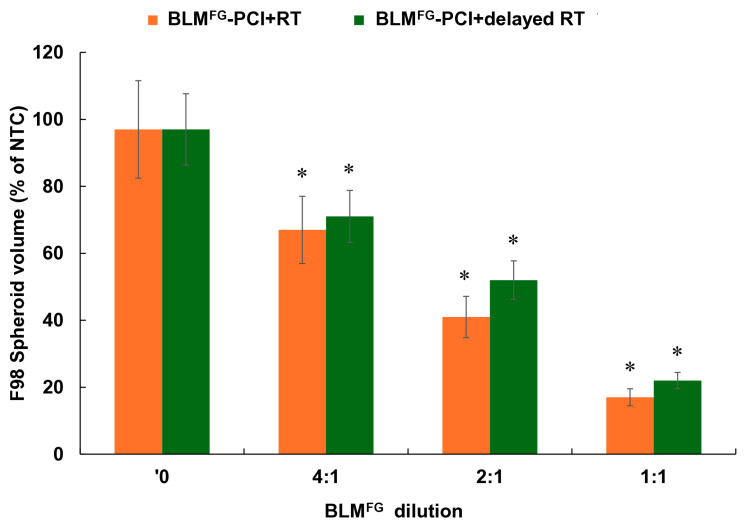
Effects of delay between BLM^FG^-PCI and RT. Spheroids received RT immediately after BLM^FG^-PCI or 24 h post-BLM^FG^-PCI. Data points represent the average volumes of spheroids after 14 days in culture, represented as a % of the non-treatment control spheroid volumes. Error bars indicate standard deviation, and * represents significant differences (*p* < 0.05) when compared to the controls.

**Figure 8 cancers-16-04029-f008:**
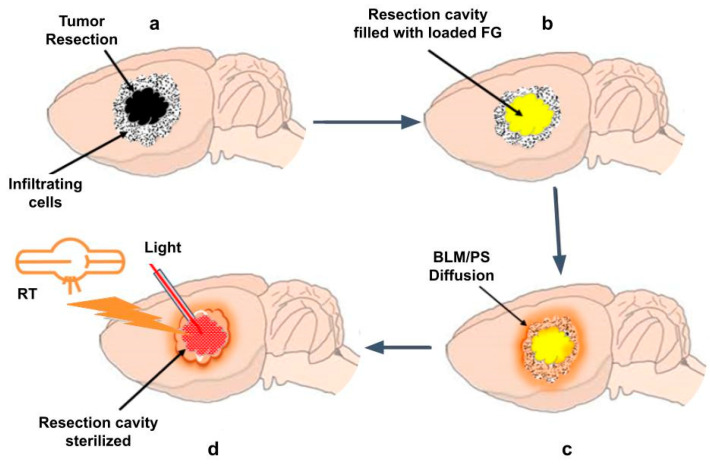
Light and RT administration to residual tumor cells in a tumor resection cavity. (**a**) Residual tumor cells in a tumor resection cavity. (**b**) Resection cavity implanted with FG loaded with BLM/photosensitizer. (**c**) BLM/photosensitizer released from FG diffuses into tumor sites. (**d**) Administration of RT by external beam and light by an implanted fiberoptic catheter in the tumor resection cavity.

**Table 1 cancers-16-04029-t001:** Synergistic effects of BLM-PCI and RT.

F98 Spheroids			
BLM	Radiation		
	6 Gy	8 Gy	10 Gy
Free BLM	1.05 ± 0.11 *	1.07 ± 0.12	1.04 ± 0.14
Free BLM-PCI	1.2 ± 0.14	4.9 ± 0.30	5.6 ± 0.36
BLM^FG^	1.15 ± 0.12	1.46 ± 0.11	1.3 ± 0.25
BLM^FG^-PCI	1.85 ± 0.14	5.0 ± 0.39	4.9 ± 0.29
**BT_4_C Spheroids**			
**BLM**		**Radiation**	
	**6 Gy**	**8 Gy**	**10 Gy**
Free BLM	1.04 ± 0.11 *	0.98 ± 0.10	1.06 ± 0.14
Free BLM-PCI	1.1 ± 0.14	4.28 ± 0.24	5.21 ± 0.33
BLM^FG^	1.01 ± 0.11	1.05 ± 0.14	1.28 ± 0.15
BLM^FG^-PCI	1.77 ± 0.12	4.83 ± 0.36	5.45 ± 0.29

* α value.

## Data Availability

All data supporting the findings of this study are available within the paper.

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
