# Peer review of "Enhancing the Efficacy of Radiation Therapy by Photochemical Internalization of Fibrin-Hydrogel-Delivered Bleomycin"

_cancers, 2024, doi:10.3390/cancers16234029_

Round 1
Reviewer 1 Report
Comments and Suggestions for Authors
The study by Laurel et al. reported an interesting approach for enhancing radiotherapy efficacy in the GBM in vitro model through incorporating photochemical internalization of fibrin hydrogel delivered bleomycin. The presented data appears enough to support the discussion. The following comments should be addressed:
1- Please show significant alterations/levels in the figures.
2- The authors did not provide direct evidence regarding this notion in conclusion: “The results of the present study show that active non-degraded BLM was released from FG layers in a prolonged manner, measured in days.”
3- Please provide suitable references for each described evidence In the manuscript:
For example, the following evidences remained without references:
“Hydrophilic drugs have a significantly greater loading ability into hydrogels compared to hydrophobic drugs. On the other hand, many highly effective chemotherapeutic agents are large and water-soluble, and therefore do not easily penetrate plasma membranes but are actively transported into cells by endocytosis. Their poor ability to escape from the resulting intracellular endosomes leads to their inactivation.”
“PCI is based on the use of photosensitizers, which localize in the cell membrane and are carried into the cell, covering the inner leaflet of the endosomal membranes. The photosensitizer remains in the endosome membrane while the macromolecule remains localized within the lumen. Specific amphiphilic photosensitizers like Al phthalocyanine desulphonated (AlPcS2a) and meso-tetraphenyl chlorin desulphonated (TPCS2a) preferentially accumulate in the membranes of endosomes and lysosomes. Upon light exposure, the photosensitizer interacts with ambient oxygen to produce singlet oxygen, which ruptures the vesicular membrane. Therefore, the released agent can exert its full biological activity, contrary to being degraded by lysosomal hydrolases following endo-lysosomal fusion.”
“Since radiation therapy forms one of the main modalities for the treatment of GBM, improving its efficacy while reducing its neuro-toxicity is an important goal. One direction, mainly through advances in high-precision technologies, has achieved significant advances in local dose escalation without undue toxicities. Techniques, such as linear accelerators with motion-tracking and beam modulation, inverse planning-based intensity modulation, as well as Gamma Knife, Cyberknife, and Tomo-therapy, have greatly contributed to this end.”
Author Response
Dear reviewer,
We sincerely appreciate your thoughtful review of our manuscript, "Enhancing the efficacy of radiation therapy by photochemical internalization of fibrin hydrogel delivered bleomycin", which we recently submitted to Cancers. Your time and effort in offering constructive feedback have been invaluable in enhancing our work. We have carefully considered your comments and made revisions, accordingly, resulting in an improved manuscript.
Thank you once again for your valuable contribution to this peer-review process.
Sincerely,
Sophia Renee Laurel
Beckman Laser Institute
University of California, Irvine
Comments 1: Please show significant alterations/levels in the figures.
Response 1: We are not sure what the reviewer is asking. Significant findings are indicated with an asterisk in the figures. The asterisk markings and stated p-value thresholds may be found in the figures and figure legends of figures 2a, 2b, 5a, 5b, and 7 – pages 5, 9 and 11.
Comments 2: The authors did not provide direct evidence regarding this notion in conclusion: “The results of the present study show that active non-degraded BLM was released from FG layers in a prolonged manner, measured in days.”
Response 2: The reviewer is correct; we have not shown the release kinetics of BLM in this paper. As stated in lines 185-187 this was determined in a previous study included as reference 13. The conclusion has been revised in accordance: “The results of the present study show that active non-degraded BLM was released from FG layers” (lines 350-351).
Comments 3: Please provide suitable references for each described evidence in the manuscript.
Response 3: Agree. The additional references have been added – References 14, 15, 17, 18, 19, 25, 30, and 31. References were added to the indicated evidence in lines 60, 67, and 279.
Reviewer 2 Report
Comments and Suggestions for Authors
The present study examined the ability of photochemical internalization of anticancer drug bleomycin relased by a combined thrombin and fibrinogen hydrogel to enhance the effects of radiation treatments on glioma spheroids in vitro. The results appear to show that bleomycin released from the hydrogel does not deteriorate and remains capable of performing its antitumor activity, and that the photochemical internalization (PCI) of bleomycin amplifies the anti-proliferative effects of radiation in comparison with the half-treatment without PCI. This procedure also reduces the radiation dose required to decrease spheroid growth. The authors hypothesize the use of this slow-release hydrogel for the treatment of post-operative recurrences after glioma resection surgery. In general, the objectives are quite clear and the comments and conclusions made by the authors are reflected in the results obtained, however, there are some questions to be clarified.
Major Revision
· There are no controls in any experiment where only AlPcS2a is present. Is there any evidence that this compound may not have antiproliferative effects in treated cells?
· In the Results section in paragraph 164-166 it says “RT alone, however, did produce a growth delay compared to NTC at 8 and 10 Gy with RT-treated spheroids reaching only 56% of NTC final volume on day 9 compared to 92% for control spheroids” but data are present. This phenomenon is clearly visible and well described for the 8 Gy treatment in figure 3 but not for 10 Gy. Is there any data to confirm this statement for 10 Gy?
· Assuming that the experiment described in section 3.5 was not done with the BLMFG but only with the free compounds is correct? Since the main aim of the work is not only to demonstrate the efficacy of the combined treatment (PCI+RT) but also to verify that the compounds released by the FG do not lose efficacy, having the results of BFGFG and BFGFG-PCI (both loaded and harvested from FG) available would be desirable. Furthermore, it is not specified on which cell line the experiment was performed.
Minor Revision
· In section 2.4 of the methods, it is not clear when the spheroids are used for experiments. Are they used 24h after their plating on 96 wells or when they reach 0.2 mm in diameter? Please clarify the conditions.
· For a more immediate understanding of the graphs, figures 3-4-5 should indicate the type of cells examined (e.g. F98/BT4C Spheroid Volume (% of NTC) on the y-axis).
· In section 3.4 in lines 226-227 it is stated that 0.12μg AlPcS2a was added to the Fibrin Glue but it is not explained why. In the case of bleomycin, in the discussion section the authors explain very well why they load the gel with twice the concentration that is then used in the experiments (lines 324-332). A brief mention also for AlPcS2a would also be appreciated.
· In section 3.5 why were different dilutions of BLMFG tested? Was a dose response intended? Please clarify.
· In section 3.6 the data for BT4C was not shown, however, it would be better to include them in the table or supplementary data.
· Check the acronym BLMFG. Especially in the final part of the paper it is spelled differently (FGBLM).
· Some sentences need references: line 63-64 “Their poor ability to escape from the resulting intracellular endosomes leads to their inactivation” and line 289-292 “Techniques, such as linear accelerators with motion-tracking and beam modulation, inverse planning-based intensity modulation, as well as Gamma Knife, Cyberknife, and Tomo-therapy, have greatly contributed to this end.”
· The Acronym SN in line 261 should be explained.
Author Response
Dear reviewer,
We sincerely appreciate your thoughtful review of our manuscript, "Enhancing the efficacy of radiation therapy by photochemical internalization of fibrin hydrogel delivered bleomycin", which we recently submitted to Cancers. Your time and effort in offering constructive feedback have been invaluable in enhancing our work. We have carefully considered your comments and made revisions, accordingly, resulting in an improved manuscript.
Thank you once again for your valuable contribution to this peer-review process.
Sincerely,
Sophia Renee Laurel
Beckman Laser Institute
University of California, Irvine
Comments 1: There are no controls in any experiment where only AlPcS2a,is present. Is there any evidence that this compound may not have antiproliferative effects in treated cells?
Response 1: AlPcS2a, has been used by us as well as others for PCI studies. At the concentrations used both in these studies and many others, it has proven nontoxic in the absence of light treatment (dark controls). Therefore, we believe it is unnecessary, in this study, to include controls in which spheroids are only treated with AlPcS2a.
Comments 2: In the Results section in paragraph 164-166 it says “RT alone, however, did produce a growth delay compared to NTC at 8 and 10 Gy with RT-treated spheroids reaching only 56% of NTC final volume on day 9 compared to 92% for control spheroids” but data are present. This phenomenon is clearly visible and well described for the 8 Gy treatment in figure 3 but not for 10 Gy. Is there any data to confirm this statement for 10 Gy?
Response 2: We have modified the statement to account for results with 10 Gy. Lines 164-166 “RT alone, however, did produce a growth delay compared to NTC at 8 and 10 Gy with RT-treated spheroids reaching only 56% of NTC final volume on day 9 compared to 92% for control spheroids” have been moved to lines 171 - 173 and the results for 10 Gy were added to the text, “A similar growth delay was also observed for both cell lines at RT = 10Gy, with spheroids reaching 49% of NTC final volume at day nine (data not shown)” (lines 173 – 175).
Comments 3: Assuming that the experiment described in section 3.5 was not done with the BLMFG but only with the free compounds is correct? Since the main aim of the work is not only to demonstrate the efficacy of the combined treatment (PCI+RT) but also to verify that the compounds released by the FG do not lose efficacy, having the results of BFGFG and BFGFG-PCI (both loaded and harvested from FG) available would be desirable. Furthermore, it is not specified on which cell line the experiment was performed.
Response 3: The experiments described in section 3.5 were done with both BLM and AlPcS₂a as free drug and the text has been revised to clarify this and the cell line used, F98 (lines 224-232). The main point of these experiments was to show that a significantly lower RT dose is required when combined with BLM-PCI than that needed for RT alone, to obtain an equivalent efficacy. The verification that the compounds released by the FG do not lose efficacy was demonstrated by comparing the results shown in figure 3 and 4 and table 1. We therefore felt it was unnecessary to also include BLMFG in section 3.5 since that point was already made.
Comments 4: In section 2.4 of the methods, it is not clear when the spheroids are used for experiments. Are they used 24h after their plating on 96 wells or when they reach 0.2 mm in diameter? Please clarify the conditions.
Response 4: The text was modified to clarify that spheroids were used 24 hours after plating the F98 or BT4C cells into 96-well plates: “Centrifugation occurred at 500g for 10 minutes to group the cells into a disk. Incubation of the plates occurred at 37ºC in 5% CO2 for 24 hours prior to receiving treatment” (lines 99 – 101). The statement regarding spheroids of 0.2 mm is intended to indicate what was observed of the spheroids 24 hours after plating: “The cells typically formed uniform 3-dimensional spheroids, approximately 0.2 mm in diameter” (lines 101 – 102).
Comments 5: For a more immediate understanding of the graphs, figures 3-4-5 should indicate the type of cells examined (e.g. F98/BT4C Spheroid Volume (% of NTC) on the y-axis).
Response 5: Thank you for pointing this out. We agree, and therefore made changes to figures 2a, 2b, 3a, 3b, 4a, 4b, 5a. 5b, 6a, 6b, and 7.
Comments 6: In section 3.4 in lines 226-227 it is stated that 0.12μg AlPcS2a was added to the Fibrin Glue, but it is not explained why. In the case of bleomycin, in the discussion section the authors explain very well why they load the gel with twice the concentration that is then used in the experiments (lines 324-332). A brief mention also for AlPcS2a would also be appreciated.
Response 5: We believe that there is a brief mention for why AlPcS₂a was added to the Fibrin Glue in section 3.4, in which both AlPcS₂a and BLM were added to the Fibrin Glue to best approximate in vivo conditions: “…This protocol, however, does not completely represent expected in vivo conditions, where both photosensitizer and drug will be simultaneously and continually released. Experiments were therefore performed where both BLM (0.6, 1.2, 2.4 μg) and AlPcS₂a (0.12μg) were loaded into the FG” (208-213).
Comments 7: In section 3.5 why were different dilutions of BLMFG tested? Was a dose response intended? Please clarify.
Response 7: Section 3.5 does not use varying dilutions of BLMFG, but rather a set concentration of 0.6μg/mL, as stated in lines 227 – 228.
Comments 8: In section 3.6 the data for BT4C was not shown, however, it would be better to include them in the table or supplementary data.
Response 8: Agreed. The data for BT4C is included in table 1 (page 11 - 12). We modified the statement in lines 272 - 273 regarding the BT4C cell line in reference to table 1.
Comments 9: Check the acronym BLMFG. Especially in the final part of the paper it is spelled differently (FGBLM).
Response 9: Thank you for bringing this to our attention. We made the necessary changes to maintain the notation BLMFG consistent throughout the text.
Comments 10: Some sentences need references: line 63-64 “Their poor ability to escape from the resulting intracellular endosomes leads to their inactivation” and line 289-292 “Techniques, such as linear accelerators with motion-tracking and beam modulation, inverse planning-based intensity modulation, as well as Gamma Knife, Cyberknife, and Tomo-therapy, have greatly contributed to this end.”
Response 10: We agree, and therefore included references 14 and 15 (line 60), and references 30 and 31 (line 344) for those sentences.
Comments 11: The Acronym SN in line 261 should be explained.
Response 11: Thank you for pointing this out. We included the term “supernatant” for the acronym SN (line 249).
Round 2
Reviewer 1 Report
Comments and Suggestions for Authors
The authors addressed all my concerns. The manuscript could be considered for publication as it is.
Reviewer 2 Report
Comments and Suggestions for Authors
To the authors,
thank you very much for accepting the comments and responding in a precise and comprehensive manner. As far as I am concerned, the paper in the present form no longer shows any issues.
Kind Regards